# THE BATCH SIZE CAN AFFECT INFERENCE RESULTS

## ABSTRACT

When performing matrix multiplication using GPUs, the cuBLAS library is commonly used for computational efficiency. Because of the cuBLAS' heuristics, a vast, deep neural network model with GPUs may produce different test results owing to the batch sizes in both the training and inference stages. In this paper, we show that the batch size affects the inference results of deep neural network models. Our test models were the well-known bidirectional encoder representations from transformers (BERT) and generative pre-trained transformer (GPT) natural language processing (NLP) models, and the super-resolution generative adversarial network (SRGAN) image generation model in FP32 and TF32. In the TF32 setting, the evaluation loss in BERT using the general language understanding evaluation (GLUE) data sometimes varied for different batch sizes. The GPT generated sentences depending on batch size, and we show the logit's mean square error by increasing the token length. The SRGAN model produced different images from batch to batch. However, these phenomena were not observed under the FP32 setting. Therefore, the batch size must be carefully managed in large-sized deep neural networks under the TF32 setting.

## 1 INTRODUCTION

Several numerical models include matrix multiplication as a fundamental component. For computational efficiency, tiling methods are employed for matrix multiplication on computers, but this leads to the accumulation of rounding errors. Considerable research has been undertaken to develop accurate and efficient matrix multiplication. Algorithm-based fault tolerance (ABFT) Kuang-Hua Huang & Abraham (1984) and autonomous-ABFT Braun et al. (2014) are two examples of representative studies. Furthermore, several algorithms have been compared in Fèvre et al. (2021). The most widely used library for performing basic linear algebra operations, such as vector addition, scalar multiplication, dot products, linear combinations, and matrix multiplication, is termed basic linear algebra subprograms (BLAS).

The BLAS is a specification for a set of low-level routines to execute standard linear algebra operations, such as vector addition, scalar multiplication, dot products, linear combinations, and matrix multiplication. They are the standard low-level routines for linear algebra libraries, including C ("CBLAS interface") and Fortran bindings ("BLAS interface"). Although the BLAS standard is broad, BLAS implementations are frequently optimized for speed on a single platform; therefore, employing them can result in significant performance gains. Most libraries that provide linear algebra functions adhere to the BLAS interface, allowing library users to create programs not connected to the BLAS library being used. With the development of GPGPU, BLAS implementations are being increasingly used, for instance, cuBLAS and rocBLAS .

Additionally, general matrix multiplication (GEMM) is the matrix multiplication method contained in BLAS. Numerous studies have been conducted to develop GEMM. For instance, the study for efficient non-square and sparse matrix computations was performed in Qin et al. (2020), the methods that fully utilize hardware resources were developed in Fatahalian et al. (2004), and research in Kelefouras et al. (2016) enhanced the optimization and effectiveness of performing GEMM on GPUs.

A deep learning model based on matrix operations is used in many diverse tasks. To train and inference a deep neural network model, it is important to compute matrices of different sizes efficiently, hence, acceleration research is being conducted to include them in various accelerators extending

from tensor cores to TPUs. A hyperscale model such as this requires a huge amount of computing resources, because of the numerous matrix computations that may reduce accuracy. Therefore, batch learning is required to overcome memory limitations, and research has been undertaken on adaptive batch sizes for successful batch learning Devarakonda et al. (2017) McCandlish et al. (2018).

However, because the batch size changes the size of the matrix to be computed, the GEMM operation method's tiling size is varied by the batch size. Large batches are often utilized for rapid training, whereas relatively small batches are used for service. In addition, there is a problem caused by the difference between the training and inference stages, and at this time, the error is not synced, thus degrading the model's performance.

Floating-point arithmetic is a method to effectively represent and calculate decimal points in computer science. It is expressed as an approximation through a floating point method at the expense of decimal-point precision and is known as a standard, such as IEEE 754. Floating-point precision for scientific operations generally uses 64-bit or more double-precision arithmetic. Because precision in deep learning has different computational costs and different precision requirements, instead of sacrificing precision, tensor operation accelerators that significantly increase the computational cost and speed are used. One representative example is NVIDIA's Tensor Core. Even when using low precision in deep learning, in certain cases, the bit configuration differs from the general floating-point standard. In general, it is composed by changing the combination of the sign bits, exponent bits, and mantissa bits that constitute the floating point. BF16 Kalamkar et al. (2019), which increases the dynamic range of single-precision FP16 and reduces significant figures, and TF32, which enables the dynamic range of 32 bits using the 19-bit representation supported after NVIDIA's Ampere architecture, etc. In general, deep learning algorithms may encounter the problem of gradient explosion by abandoning the dynamic range. Increasing the dynamic range by abandoning significant digits at the same precision is effective.

In this study, we used the bidirectional encoder representations from transformers (BERT) Devlin et al. (2018) and the generative pre-trained transformer (GPT) Brown et al. (2020) in natural language processing (NLP) models and the super-resolution generative adversarial network (SRGAN) Ledig et al. (2016) in an image generation model to estimate the phenomenon to investigate whether the batch size can affect the inference results under two types of floating-point arithmetic systems, TF32 and FP32. In BERT, the evaluation loss was compared using the benchmark GLUE data set Wang et al. (2018) by varying the training batch and the inference batch. In GPT, generated sentences were compared for batch size with the logit's mean square error when increasing the token length. In SRGAN, generated images were compared for inference from 256 data elements by batch to the data trained with 200,000 CelebFaces Attributes Dataset (CelebA) images Liu et al. (2015).

## 2 EXPERIMENTS

We experimentally tested the three models BERT, GPT, and SRGAN. The GPUs specifications for this experiment were NVIDIA RTX 3090 (ampere) in TF32 and NVIDIA 2080 TI (Turing) in FP32.

### 2.1 BERT

Developed by Google in 2018 , BERT is a transformer-based language model. We performed fine tuning according to the batch size using the GLUE dataset based on BERT's pre-training data and performed text classification. A batch size of 32 was used for training and evaluation loss was measured using 1, 2, 4, 8, 16, 32, 64, and 128 batches for inference. Subsequently, significant differences were found in three datasets (CoLA, RTE, SST-2) out of nine GLUE datasets (Figure 1). This phenomenon did not occur in FP32 (Figure 1).

### 2.2 GPT

GPT is an unsupervised transformer language model created by OpenAI, which translates text, answer questions, summarizes passages, and generates text output. In GPT, the inference of generating texts was conducted based on the official trained model. The input data of all batch sizes is same. We choose that the batch size of the model is 1,2,4 and 8, and the FP32 and TF32 settings of NVIDIA are activated to enable computation acceleration of the tensor core. Figure 2 consists of the same

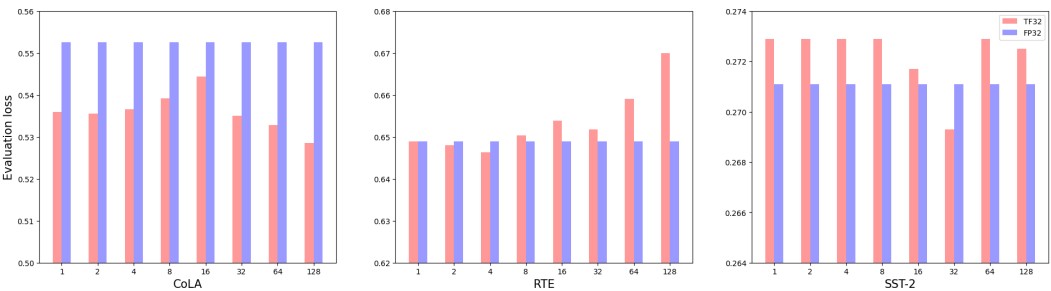

Figure 1: Evaluation loss according to batches in TF32 and FP32 settings

input as the different batch used for the inference. And the mean square error(MSE) of the output token logit was shown. From these experimental results, it can be said that the logit value, that is, the probability value of the next token generated by iterative calculation of the GPT model, has unstable numerical calculation consistency in precision in TF32 compared to FP32. Among them, the forward calculation with different batch sizes while using TF32 precision has a relatively large error. In fact, most deep learning algorithms learn and evaluate by placing data through tensors. However, when inferring the model, most use a different batch size than the training time. This effect appears larger as the model is larger and the number of calculations increases, and it can be easily confirmed in a model with a large amount of iterative computation such as GPT.

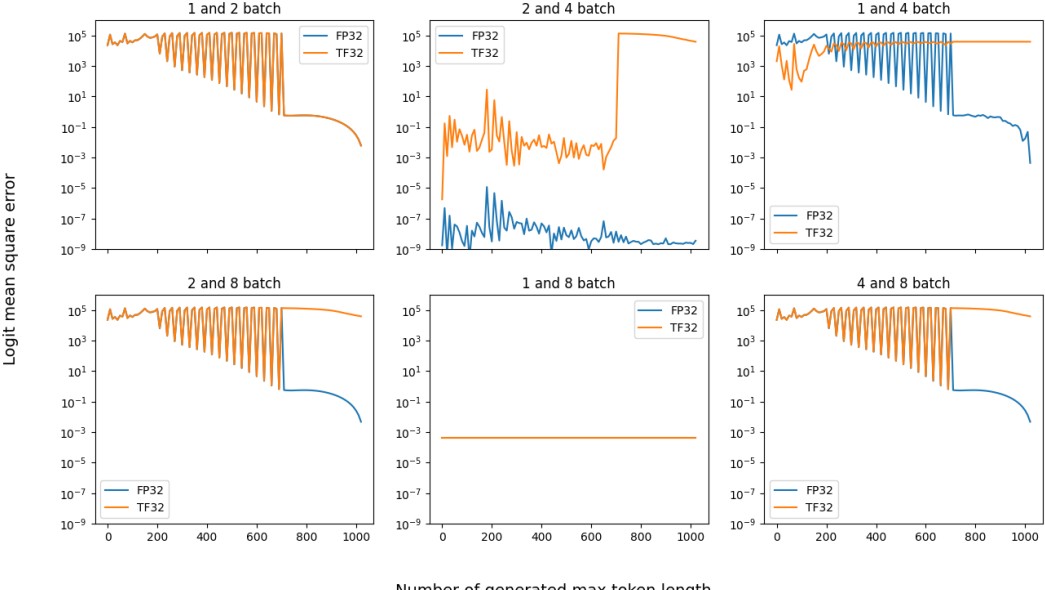

Figure 2: Mean square error between inference logit values of different batches when increasing token length

## 2.3 SRGAN

SRGAN is a generative adversarial network (GAN) for image super-resolution (SR). For SRGAN training, around 200,000 CelebA images were trained in 16 batches, and 4X high-resolution image output, which is the basic size of SRGAN was used. Therefore, the size of 218*178 becomes SR at 872*712. The inferences stages for the trained model were 1, 2, 4, 8, and 16 batches. In FP32, all generated images were identical regardless of the batch size. In contrast, in TF32, the images were not identical when using 1 and 2 batches, 4 and 8 batches, and 16 batch while the images with 1 batch and 2 batch and those with 4 batch and 8 batch were identical, respectively (Table 1).

Table 1: Mean number of non-identical pixels in 256 generated images with different batch size in TF32

| Batch size | 1 | 2 | 4 | 8 | 16 |
|---|---|---|---|---|---|
| 1 | 0 | 0 | 121.9 | 121.9 | 123.0 |
| 2 | | 0 | 121.9 | 121.9 | 123.0 |
| 4 | | | 0 | 0 | 22.9 |
| 8 | | | | 0 | 22.9 |
| 16 | | | | | 0 |

## 3 DISCUSSION

Our experiments show that the results of GEMM and floating-point operations in deep learning problems may compute different results from the training and inference batches, and this is due to swapping, which accumulates errors owing to matrix operations and floating-point numbers. This problem may appear in various domains, such as the text and image fields, but it is not recognized as a problem because the known models are not large enough and the numerical precision used in the training is not well disclosed. In addition, in a general generative model, it is difficult to construct a metric to measure the quality of the generative model, and there are almost no studies on performance being reduced by the batch size at the time of inference, which is not a part of learning. In the case of text, owing to the presence of a random part of the generative model, the part where the corresponding effect is not well recognized also has considerable influence. In image generation models, such as SRGAN, image quality may be affected if continuous images are processed or the batching is varied. Therefore, when performing GEMM operations for matrix operation on GPUs, the batch sizes should be carefully managed in large-sized deep neural networks.

## 4 CONCLUSION

We have experimentally shown that errors occur as the batch sizes for training and inference vary in a deep learning model. This error can affect the model's performance(not performance but stale?). This problem is caused as the GEMM operation on the GPU changes heuristically according to the size of the input matrix. Owing to the characteristics of the GPU hardware, this variable calculation by GEMM is essential for optimizing the operation speed; thus, if it is forcibly controlled, the speed will inevitably decrease.

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
