# OpenReview forum: "The batch size can affect inference results"
_ICLR.cc/2023/Conference — Submitted to ICLR 2023_

### Official Review · Reviewer_YAmr · 2022-10-20

**Confidence:** 3
**Correctness:** 4
**Technical Novelty And Significance:** 3
**Empirical Novelty And Significance:** 3
**Recommendation:** 3

**Clarity, Quality, Novelty And Reproducibility:**

Clarity: good

Quality and Novelty

Overall I am afraid this paper is lack of strong innovation and understanding. It only presents some experiment results, while did not offer sufficient understanding on why/how it happened.

For example, here are some questions:
In figure 1 (a) I am curious to see in CoLA TF32 has much lower loss compared to TF32 on all batch sizes. Is it expected?
In figure 2, why the patterns for 1 vs 2, 2 vs 8, and 4 vs 8 are so similar, while the others are so different?
Continue my last question, in the 1 vs 2, 2 vs 8, and 4 vs 8  there are clear systematic on error vs max length, is there any understanding why it happened?
In table 1 it shows the number of non-identical pixels. I am curious how such much those pixels different? It would be helpful to also add example figures for comparison.

Reproducibility:
Most of the experiment settings are clear, though I was not able to find the code so I am not confident how easy to reproduce the experiment.
For example, authors mentioned "A batch size of 32 was used for training", so which GPU specifications is used for model training? I assume in evaluation the model is the same for both FP32 and TF32?


**Details Of Ethics Concerns:**

No concerns

**Strength And Weaknesses:**

Overall the topic on the impact of batch size on the performance is interesting. However, this paper lack of sufficient deep understanding on how it happens and how to prevent it.

**Summary Of The Paper:**

In this work, authors show that the batch size affects the inference results of deep neural network models.

In the empirical study, authors studied bidirectional encoder representations from transformers (BERT) and generative pre-trained transformer (GPT) natural language processing (NLP) models, and the super-resolution generative adversarial network (SRGAN) image generation model in FP32 and TF32.

The study shows batch size will influence TF32 setting while not observed under the FP32 setting.

**Summary Of The Review:**

Overall I am afraid the depth in this paper is not sufficient for a long paper in ICLR. It may better fit toas a short workshop paper.

---

### Official Review · Reviewer_ReKx · 2022-10-24

**Confidence:** 4
**Correctness:** 2
**Technical Novelty And Significance:** 1
**Empirical Novelty And Significance:** 1
**Recommendation:** 1

**Clarity, Quality, Novelty And Reproducibility:**

The paper was extremely hard to follow, between grammatical errors, sentences with no verb and acronyms not properly introduced. A few samples that I did not understand:
- “Because of the cuBLAS’ heuristics, a vast, deep neural network model with GPUs may produce different test results owing to the batch sizes in both the training and inference stages.”
- “The GPT generated sentences depending on batch size, and we show the logit’s mean square error by increasing the token length.”
- “The SRGAN model produced different images from batch to batch”
- “In addition, there is a problem caused by the difference between the training and inference stages,”
- “BF16 Kalamkar et al. (2019), which increases the dynamic range of single-precision FP16 and reduces significant figures, and TF32, which enables the dynamic range of 32 bits using the 19-bit representation supported after NVIDIA’s Ampere architecture, etc”
- Figure 2: “number of generated max token length”
- Figure 1: what is the sequence length used?
- What does “1 and 2 batch” mean? For instance Figure 2.
- …


**Strength And Weaknesses:**

Strengths:
- It seems like an interesting problem and I don’t think people are aware enough of this behavior.

Weaknesses:
- The paper is mostly about shedding light on the discrepancy during inference (could not tell whether the training case was studied). For this study to be interesting, I would have expected some deeper analyses of why such discrepancy is happening, if inference batch sizes need to match training batch sizes, if not to what discrepancy is acceptable, formulate recommendations, etc.
- The paper is very hard to follow, and I had a hard time putting things together. For instance, I am still unclear on most of the experiment steups.


**Summary Of The Paper:**

This paper sheds light on discrepancies in results when using tf32: depending on the batch size used for inference (and training?), the same instance will give different results (as measured as logits, pixel values or loss depending on the setup).

**Summary Of The Review:**

This submission is extremely incomplete and does not meet the standard for a publication venue such as ICLR.
It seems like the paper is describing the very first step of a project: verifying and exposing the problem. Beyond that, the contributions are non-existent.

---

### Official Review · Reviewer_7wfT · 2022-10-25

**Confidence:** 5
**Clarity, Quality, Novelty And Reproducibility:** The work is not ready to be accepted …
**Correctness:** 3
**Technical Novelty And Significance:** 3
**Empirical Novelty And Significance:** 3
**Recommendation:** 3

**Strength And Weaknesses:**

An interesting finding! It experimentally finds that the different batch sizes during training and inference will affect the model performance due to the matrix operation on GPU. Therefor the authors suggest to carefully choose the batch size.

However,
1. The paper is not well-written in presentation. The format of the reference is not correct. Table 1 lacks the annotation of the lines and columns. In Figure 2, it is also not clear what the '1 and 2 batch' means. The expression of "batch/batches" is also confusing. Do the authors mean batch size? 4 batch and 4 batch size refer to different meanings.
2. The work is not ready in its current form to be published at the ICLR conference with only very limited experimental results without more deep analysis.

**Summary Of The Paper:**

This paper experimentally shows that errors occur as the batch sizes for training and inference vary in
a deep learning model by caused as the GEMM operation on the GPU.

**Summary Of The Review:**

The work has some interesting findings through experiments. But it is not ready to be published at its current form without any deeper analysis.

---

### Decision · Program_Chairs · 2023-01-20

**Decision:**

Reject

**Justification For Why Not Higher Score:**

Clear rejection

**Justification For Why Not Lower Score:**

Clear rejection

**Metareview: Summary, Strengths And Weaknesses:**

The paper empirically shows that the batch size can affect inference results. However, the paper is not well prepared and lacks substance (only 3.5 pages)